# Affinity Proteomics Identifies Interaction Partners and Defines Novel Insights into the Function of the Adhesion GPCR VLGR1/ADGRV1

**DOI:** 10.3390/molecules27103108

**Published:** 2022-05-12

**Authors:** Barbara Knapp, Jens Roedig, Heiko Roedig, Jacek Krzysko, Nicola Horn, Baran E. Güler, Deva Krupakar Kusuluri, Adem Yildirim, Karsten Boldt, Marius Ueffing, Ines Liebscher, Uwe Wolfrum

**Affiliations:** 1Institute of Molecular Physiology (ImP), Molecular Cell Biology, Johannes Gutenberg University Mainz, 55128 Mainz, Germany; bknapp@web.de (B.K.); jens.roedig@live.de (J.R.); heiko.roedig@live.de (H.R.); jkrzysko@uni-mainz.de (J.K.); bgueler@uni-mainz.de (B.E.G.); deva.kusuluri@unige.ch (D.K.K.); yildadem@uni-mainz.de (A.Y.); 2Core Facility for Medical Bioanalytics, Institute for Ophthalmic Research, University of Tuebingen, 72076 Tuebingen, Germany; nicola.horn@medizin.uni-tuebingen.de (N.H.); karsten.boldt@uni-tuebingen.de (K.B.); marius.ueffing@uni-tuebingen.de (M.U.); 3Rudolf Schönheimer Institute of Biochemistry, Faculty of Medicine, Leipzig University, 04103 Leipzig, Germany; ines.liebscher@medizin.uni-leipzig.de

**Keywords:** adhesions GPCR, Usher syndrome, affinity proteomics, protein networks

## Abstract

The very large G-protein-coupled receptor 1 (VLGR1/ADGRV1) is the largest member of the adhesion G-protein-coupled receptor (ADGR) family. Mutations in VLGR1/ADGRV1 cause human Usher syndrome (USH), a form of hereditary deaf-blindness, and have been additionally linked to epilepsy. In the absence of tangible knowledge of the molecular function and signaling of VLGR1, the pathomechanisms underlying the development of these diseases are still unknown. Our study aimed to identify novel, previously unknown protein networks associated with VLGR1 in order to describe new functional cellular modules of this receptor. Using affinity proteomics, we have identified numerous new potential binding partners and ligands of VLGR1. Tandem affinity purification hits were functionally grouped based on their Gene Ontology terms and associated with functional cellular modules indicative of functions of VLGR1 in transcriptional regulation, splicing, cell cycle regulation, ciliogenesis, cell adhesion, neuronal development, and retinal maintenance. In addition, we validated the identified protein interactions and pathways in vitro and in situ. Our data provided new insights into possible functions of VLGR1, related to the development of USH and epilepsy, and also suggest a possible role in the development of other neuronal diseases such as Alzheimer’s disease.

## 1. Introduction

The longest isoform 1b of the very large G-protein-coupled receptor 1 (VLGR1/ADGRV1), also known as GPR98, MASS1, or FEB4, is the largest adhesion G-protein-coupled receptor (ADGR) in the human body (Figure 1a) [1,2,3]. Mutations in *VLGR1/ADGRV1* can manifest in two disease phenotypes. Most VLGR1 mutations are causative for the human Usher syndrome type 2C (USH2C) [4]. Usher syndrome (USH) is a severe genetically heterogenous autosomal recessive disorder and the most common cause of hereditary deaf-blindness [5,6]. There is growing evidence for a relation of haploinsufficiency of VLGR1 to epilepsy in humans [7,8,9,10]. This association with epilepsy is consistent with the audiogenic seizure phenotype in VLGR1 mutant mice [11,12,13].

VLGR1 is mainly expressed in the nervous system, with a strong expression in the eye and the inner ear [5,12,14,15,16]. In both sensory organs, the VLGR1 protein is found at the synapses of the sensory cells [17,18]. In inner ear hair cells, VLGR1 is an essential component of the ankle links, which connect neighboring stereocilia, and thereby stabilizes the nascent hair bundles of the cochlear during differentiation [14,16,19]. In the retinal photoreceptor cells, VLGR1 is also associated with fibrous connectors [20]. Here, the extremely long adhesive extracellular domain of VLGR1 spans between the membranes of the apical inner segment and of the connecting cilium of photoreceptor cells. In mice, defects in VLGR1 cause the disruption of the fibrous links and the membrane–membrane adhesion in both sensory cell types [14,20].

Similar to other ADGRs, VLGR1 possesses a unique structure (Figure 1a) characterized by the N-terminal extracellular domains (ECD) that mediate cell–cell or cell–matrix adhesion, combined with a secretin-like GPCR moiety [3] that includes a seven transmembrane (7TM) domain and a cytoplasmic region. It is the N-terminus that mainly contributes to the enormous size of VLGR1, which can undergo alternative splicing, thus yielding also shorter variants such as VLGR1a (Figure 1) [2]. An important feature of ADGRs is their capability to be self-cleaved at the so-called G-protein-coupled receptor proteolysis site (GPS) [21], located close to the first of the seven transmembrane (7TM) domain and embedded in the GPCR autoproteolysis-inducing domain (GAIN) [22]. The two resulting fragments, an N-terminal (NTF) and a C-terminal fragment (CTF), mostly stay associated as a dimer. However, there is increasing evidence that the NTF and CTF can execute independent functions [23,24]. In particular, the expression of the NTF mediates a signal independent from the coupled G protein, while the CTF alone results in the increased activation of downstream signaling pathways compared to the full-length protein [25,26,27,28]. This is due to a tethered internal agonist that comprises the first 5–10 amino acids of the CTF, the so-called *Stachel* [29]. For some ADGRs, there is evidence that the release of the NTF triggers the binding of the *Stachel* peptide to the exoplasmic face of the ADGRs and, in consequence, induces the conformational changes, which allows the binding of ADGR effector proteins. In addition, there is also evidence for several ways of *Stachel*-independent receptor stimulation, proving that ADGR activation is complex and far away from being fully understood [30].

In any case, the active and inactive receptors most probably associate with different protein complexes enrolled in diverse functional modules. Previous analyses of the protein–protein interaction revealed that the cytoplasmic tail of VLGR1 binds through its PDZ-binding motive to the PDZ domains of USH-related scaffold proteins, namely whirlin (USH2D), harmonin (USH1C), and the USH genetic modifier PDZD7 [13]. These interactions also mediate the integration of VLGR1 into the USH-protein networks associated with membrane adhesion complexes. More recently, it became evident that defects in USH molecules lead to dysfunctions in primary cilia, which are regarded as the sensory antennae and signaling hub of the cell [31,32,33,34]. However, nothing is known about VLGR1 signaling in primary cilia [35]. In addition, two studies addressing VLGR1 signaling pathways reported contradictory pathways via Gα_s_/Gα_q_ and Gα_i_, respectively [36,37].

Here, we aimed to identify novel proteins interacting with VLGR1 and thereby define signaling pathways, protein networks, and functional cell modules related to VLGR1. For this, we applied tandem affinity purifications (TAPs) coupled to mass spectrometry analysis using different fragments of VLGR1 as bait. We identified numerous putative novel complex partners, that we analyzed by bioinformatics tools to group the identified prey according to their GO terms, and preliminarily validated some of the interactions. Our analysis identifies VLGR1-related cell modules which shed light on possible cellular functions of VLGR1 including neural development, gene and cell cycle regulation, cell adhesion, and photoreceptor function.

## 2. Results and Discussion

### 2.1. Identification of Novel VLGR1 Protein Complex Partners by Tandem Affinity Purification (TAP)

To identify novel interactors of VLGR1 we applied affinity proteomics using TAPs [38]. We fused the tandem Strep II-FLAG (SF)-tag to the N-terminus of a full-length VLGR1a isoform, the N- and/or C-terminus of the C-terminal fragment (VLGR1_CTF), and the end of the C-terminus of the intracellular domain of VLGR1 (VLGR1_ICD) (Figure 1b). These SF-tagged baits were expressed in HEK293T cells and TAPs were performed. Eluted protein complexes were separated by liquid chromatography and their peptide content determined with tandem mass spectrometry (LC-MS/MS). To identify interacting proteins, we searched the raw spectra against SwissProt databases and verified the results applying the Scaffold program. All prey identified for the four different VLGR1 constructs as baits are listed in the Appendix A.

We identified ~1000 novel putative interacting partners of VLGR1 in TAPs with both VLGR1_CTFs and VLGR1a as baits. In contrast, only 40 hits were found in VLGR1_ICD CT TAPs (Figure 1c). Half of the molecules identified in VLGR1_ICD TAPs were also found in TAPs with the longer constructs VLGR1_CTFs and VLGR1a (Figure 1d). Comparisons among TAP data sets between the two VLGR1_CTFs and VLGR1a revealed an intersection of over 500 (Figure 1e). However, our analyses also revealed high diversity in prey found in the different TAPs. The putative interaction partners belong to very heterogeneous protein groups and families that exert functions in diverse cellular compartments. This may reflect diverse functions of VLGR1 in different cellular contexts and/or may also depend on the activation state of VLGR1. This section may be divided by subheadings. It should provide a concise and precise description of the experimental results, their interpretation, as well as the experimental conclusions that can be drawn.

#### 2.1.1. Specificity of TAP Hits

In order to gauge the reliability of the identified interaction partners, we compared our datasets to the *Contaminant Repository for Affinity Purifications*, short CRAPome, which contains a collection of common contaminants in affinity proteomics data sets [39]. About half of the identified VLGR1 prey occur in less than 5% of all protein listed in the CRAPome (corresponding to a cutout ≤20), which indicates a high degree of specificity.

A comparison of TAP hits with previously identified VLGR1 binding partners [35] confirmed the two USH scaffold proteins whirlin (DFNB31/USH2D) and harmonin (USH1C) as the binary binding partners of VLGR1 in VLGR1_ICD TAPs (Appendix A). We have also previously shown that VLGR1 directly binds with its PDZ-binding motif (PBM) in the C-terminal end of the VLGR1_ICD to one of the PDZ domains of the two scaffold proteins [15,17]. Although there is no doubt that we have also identified some false-positive interactors, our validation proves the specificity of selected interactions and confirms the suitability of our affinity capture approach for the identification of novel interactors of transmembrane receptor proteins, such as VLGR1.

#### 2.1.2. Grouping and Categorizing of TAP Hits

We grouped and categorized TAP hits by Gene Ontology (GO) term enrichment analyses, applying the Cytoscape (http://www.cytoscape.org, accessed on 10 September 2017) plugin ClueGO [40] for all three GO term categories, *Biological Process, Cellular Component, and Molecular Function,* and, in addition, for *KEGG (Kyoto Encyclopedia of Genes and Genomes) pathways* (Appendix A). In our data sets, we found multiple, often unexpected, links to different cellular processes. Some of the most interesting connections are examined and discussed in the following.

### 2.2. VLGR1 G Protein Coupling Switches between Gα_s_/Gα_q_ and Gα_i_

As expected for a GPCR, we identified in our VLGR1 TAPs several subunits of heterotrimeric G-proteins, namely Gβ1 (GNB1), Gβ2 (GNB2), Gαi3 (GNAI3), and Gαs (GNAS). Interestingly, we found the association of a full-length VLGR1a with Gαi and Gαs, respectively, whereas the VLGR1_CTFs only interacted with Gαi, which primarily inhibits the cAMP dependent pathway by inhibiting adenylyl cyclase activity, decreasing the production of cAMP and resulting in decreased activity of the cAMP-dependent protein kinase. To analyze the Gα coupling of VLGR1 in more depth, we first measured the concentration of the second messenger cAMP in HEK293T cells expressing VLGR1a (Figure 2a). Overexpressing VLGR1a revealed a dose-dependent increase of cAMP, indicating the coupling of VLGR1 to Gαs. We also found a significant increase of cAMP upon the co-expression of VLGR1a with a Gαsq chimera, suggesting an activation of Gαq (Figure 2b). Based on the assumption that the CTF mutant represents the active conformation of VLGR1, we expected to see an increase in cAMP levels (Gs pathway) as well as IP levels (Gq pathway) when comparing CTF to full length constructs (Figure 2c). The proper surface and total expression of the full-length and CTF mutant was controlled using ELISA assays (Figure 2d). Surprisingly, constitutive activity of the CTF was seemingly only confirmed for Gq-mediated IP accumulation, while Gs-mediated cAMP levels were reduced. However, cAMP reduction can also be caused by activation of Gi proteins. Indeed, utilizing a Gqi chimera, this activation could be shown for the CTF mutant. These findings are consistent with previous results from independent groups reporting signaling of VLGR1 via Gαs/Gαq and Gαi [36,37]. Shin et al. (2013) used a shortened version of VLGR1 that contained an artificial NTF comprising the PTX domain, the EAR domain, and five Calxα domains. This VLGR1 version appeared to be essential for Gαs signaling, which is in line with our data observed with VLGR1a (Figure 2b). In contrast, Hu and coworkers (2014) used the VLGR1_CTF for their measurements and found the coupling to Gαi. In accordance with these previous results, we observed a stronger activation of Gαi with VLGR1_CTF, compared to VLGR1a (Figure 2c). For Gαs, we did not detect significant differences between both constructs in their basal activity levels. However, the association of Gαs with VLGR1a, but not VLGR1_CTF in our TAPs, and the dose-dependent increase of basal activity upon VLGR1a overexpression (Figure 2a), all indicate a preferred coupling of Gαs to full-length VLGR1a, compared to VLGR1_CTF, which in turn favors Gi-coupling. A signaling switch from basal Gs to active Gi has been previously shown for ADGRG3/GPR97 [41]. Interestingly, switches between Gα coupling are not restricted to ADGRs, but were previously also reported for other GPCRs [42]. A physiological role for these signaling switches in VLGR1 remains to be determined. Interestingly, the activation of the Gq protein is utilized by the receptor in its full-length as well as in its CTF version (Figure 2b,c). The same Gq-preferred activation can be found when screening for potential *Stachel*-derived agonistic peptides in a cAMP assay. We found that peptides of 10 and 11 amino acids of the respective tethered agonist region could significantly activate VLGR1 when adding a Gsq chimera (Figure 2e). As in other ADGRs, the release of the NTF from VLGR1 may trigger the binding of this 11-amino-acids *Stachel* peptide to the exoplasmic face of the VLGR1_CTF. The resulting conformational changes in the receptor may induce the switch from Gαs- to Gαi-mediated signaling of VLGR1, as indicated in the TAPs for VLGR1a and VLGR1_CTF.

### 2.3. TAP Data Analysis Indicates Coupling of VLGR1 to Various Downstream Signaling Pathways

Enrichment analyses of VLGR1 TAP data sets in the categories of the KEGG pathway and Biological Process displayed the participation of VLGR1 at 13 different signaling pathways. However, the number of prey identified with three different VLGR1 baits for these pathway categories obviously differs. Based on these differences we were able to predict how VLGR1 molecules are connected to the specific pathway. KEGG pathway analysis revealed, for VLGR1_ICD prey proteins, two mitogen-activated protein kinases, MAPK1 and MAPK3, categorized to the mTOR-pathway, which is involved in the cellular response to energy levels and nutritional or environmental cues [43]. An enrichment of prey related to mTOR-signaling was also found for SF-N-VLGR1a but not for VLGR1_CTF TAPs.

Categorized pathways assigned by the identified prey proteins identified in TAPs with the VLGR1a and VLGR1_CTF as baits are listed in Table 1. For both, full-length VLGR1a and the VLGR1_CTFs, we found proteins which participate in the AMP-activated protein kinase (AMPK) pathway, MAPK signaling, the neurotrophin pathway, and Wnt signaling. Prey proteins connected to Hypoxia-inducible factor 1 (HIF-1) signaling were enriched for VLGR1a as bait. This suggests that the less active full-length VLGR1a participates in the HIF-1 pathway and regulations related to hypoxia [44]. In contrast, protein prey related to the Notch signaling pathway, the insulin receptor-mediated pathway, and Ephrin signaling or the sphingolipid pathway were almost exclusively identified in TAPs performed with VLGR1_CTFs. The interaction of these pathways with VLGR1 probably only occurs in the active receptor stage.

Interestingly, protein prey categorized to integrin signaling was mainly associated with SF-C-VLGR1_CTF, but not with SF-N-VLGR1_CTF. The N-terminal tags at VLGR1_CTF most probably may hinder receptor activation by the tethered *Stachel* agonist [29,45]. Thus, our data favor that *Stachel*-mediated activation of VLGR1 is essential for downstream integrin signaling. In contrast, proteins involved in Wnt and Sonic Hedgehog (Shh) signaling, namely Smoothened (SMO), Frizzled 7 (FZD7), and protein tyrosine kinase 7 (PTK7), were only found in TAPs with the N-terminally tagged VLGR1_CTF. This indicates that a free, untagged C-terminus of VLGR1 is important to interact with these proteins. Notably, VLGR1, SMO, and FZD7 all contain C-terminal class I PDZ-binding motifs (PBMs). The binding of PBMs to PDZ domain scaffolding proteins likely mediates the interaction of the proteins and their clustering at the membranes. Moreover, we found 30 proteins related to the ER-associated protein degradation (ERAD) pathway that is activated by misfolded proteins [46]. This might be due to overexpression of the aGPCR bait. We also noticed that VLGR1_CTFs interacted with several proteins associated with apoptotic pathways. However, we did not observe an apoptotic effect upon the overexpression of these molecules in HEK293T cells.

### 2.4. VLGR1 Participates in Signaling at Focal Adhesions

Beside their signaling function, ADGRs possess adhesive properties. In VLGR1 TAPs with HEK293T cell lysates, we identified 96 proteins that are assigned to the GO term focal adhesion, of which 25 have a CRAPome value 20 (Figure 3a). The high number of focal adhesions (FA)-associated prey is consistent with data from RPE1 cells performed in a parallel study [47]. Figure 3b shows the localization of VLGR1 in FAs of primary mouse brain astrocytes, concordant with data of the latter study [47]. FAs are macromolecular assemblies at contact sites of cell membranes with the extracellular matrix (ECM) [48]. There they act as hubs for bidirectional signaling, the ‘‘inside-out’’ transmission of intracellular forces to the ECM, and the ‘‘outside-in’’ signal transmission of shear forces between the cell and the ECM from the environment to the cell interior [49]. While no FA protein was found in the VLGR1_ICD TAPs, the prey for SF-N-VLGR1a, SF-N-VLGR1_CTF, and SF-C-VLGR1_CTF contained several FA core proteins. Being proteins directly associated with integrins, the core FA adhesion molecules were mostly restricted to both VLGR1_CTF TAPs. This suggests that the interaction with integrin complexes depends on the activity level of VLGR1. All in all, these data further support our recent findings that VLGR1 functions as a metabotropic mechanosensor in FAs that is critical for cell spreading and cell migration [47].

### 2.5. VLGR1 Partners Participate in Cell Cycle Regulation, Cell Division, and Ciliogenesis

Our TAP data showed a high enrichment for proteins attributed with the GO term cell cycle (over 100 prey). VLGR1 interaction partners with a CRAPome value ≤ 20 could be further grouped into positive cell cycle regulators, negative cell cycle regulators, and other proteins (Figure 4a). It is worth noting that APP and other Alzheimer’s disease-related proteins, such as APBB1 and ADAM17, are part of these categories. In addition, we identified the checkpoint kinases, cell division cycle 6 (CDC6), CDC7, and CDC45, which regulate G1/S transition [50,51,52].

We also observed that over 30 proteins were additionally assigned with the GO terms spindle, and, more specifically, the spindle pole (Figure 4b). The interaction partners for the full-length VLGR1a and VLGR1_CTF partially differed in these categories, suggesting different roles of the full-length protein and its C-terminal fragment in this cellular context. The association of VLGR1 with the spindle apparatus was confirmed by the immunocytochemical staining of spindle poles by antibodies raised against the C-terminal part of VLGR1 in dividing RPE1 cells (Figure 4c). In contrast, VLGR1 was localized to the base of primary cilia in quiescent cells (Figure 4d), which is in line with the localization of VLGR1 at the periciliary membrane complex of photoreceptor cells [20]. The SiRNA-mediated knock-down of VLGR1 revealed a significant reduction in cilia length when compared to the non-targeting control (NTC) knock-down (Figure 4e). This fits to our finding that various novel identified interaction partners of VLGR1, such as nucleotide binding protein 2 (NUBP2), the IQ motif containing B1 (IQCB1/NPHP5 (nephrocystin-5)), the actin-binding protein Filamin A (FLNA), and Fas binding factor 1 (FBF1), participate in ciliogenesis [53,54,55,56]. FBF1 and Filamin A both play a role in the modification, positioning, and anchoring of the mother centriole at the plasma membrane during ciliogenesis, and IQCB1/NPHP5 is thought to additionally regulate the ciliary import at the ciliary base of the cilium where VLGR1 is found [54,57].

Furthermore, we identified proteins which have been previously associated with non- and syndromic retinal ciliopathies: the Retinitis pigmentosa 1 protein (RP1), localized in the axoneme of photoreceptor cilia [33,58], and the Kinesin family member KIF11, which is found in the periciliary region at the base of the photoreceptor cilium [59], but interestingly, also at the spindle poles of dividing cells [60]. In summary, we found evidence for the involvement of VLGR1 in cell cycle regulation and ciliogenesis. The molecular background of these functions remains to be determined.

### 2.6. VLGR1 Is Part of Protein Networks in the ER and Nucleus

Our TAP datasets contain over 400 proteins, which are assigned to the Cellular Component GO term nucleus. The vast majority of these proteins also belong to the GO term category of the nuclear outer membrane–endoplasmic reticulum membrane network. About 196 proteins are part of the latter group, including 124 proteins with a CRAPome value ≤ 20, which suggests a high specificity. For the GO term category “endoplasmic reticulum”, a large number of prey were found for SF-C-VLGR1_CTF (334), SF-N-VLGR1_CTF (348), and VLGR1a (259). In contrast, in SF-N-VLGR1_ICD TAPs, only two proteins were categorized to the ER. The association of VLGR1 with the nucleus suggested by the GO terms of the prey proteins were confirmed by the immunocytochemistry in the primary astrocytes derived from a murine brain. Immunostaining demonstrated the localization of VLGR1 in the plasma membrane (arrowhead), nucleus, focal adhesions (arrow), and in the perinuclear cytoplasm (asterisk) (Figure 5a). The association of VLGR1 with the ER suggested by the GO terms was verified by immunostaining VLGR1 and the ER component CLIMP63 in HeLa cells (Figure 5b). The staining of VLGR1 overlapped not only with the nuclear marker DAPI, but also with the ER marker CLIMP63, indicating the presence of VLGR1 in the ER. There, as with other transmembrane receptors, the aGPCRs are properly folded by chaperones present in the ER to be transported to the site of their activity. In addition, there is also evidence from our TAPs that VLGR1 may well play a role in ER function. In several TAPs, the prey were assigned to mitochondria-associated membranes (MAMs), a compartment that is formed by the adhesion of the ER membrane to the outer mitochondrial membrane [61]. This is in line with the localization of VLGR1 in this ER–mitochondria junction complex, as perilously shown [62]. In a parallel study, we have demonstrated that VLGR1 is part of a protein complex which is essential for the proper release of Ca^2+^ from the ER of the MAM, thereby controlling the Ca^2+^ homeostasis (Krzysko et al., in prep).

The identified prey of the nuclear categories participate in nuclear-specific functions, such as gene regulation, pre-RNA splicing, and transcription. We found 20 proteins that fall into the transcriptional factors and regulators category, and which have a CRAPome value ≤ 20 (Figure 5c). These include the ligand-dependent nuclear receptor interacting factor 1 (LRIF1), a repressor of retinoic acid receptor alpha (RARA) transcriptional activity [63]. Further, we identified transcriptional regulators that are involved in brain and inner ear development (FOXG1) [64,65], and the differentiation of progenitor cells in the retina (NR2F1) [66]. Moreover, amyloid beta precursor protein binding family B member 1 (APBB1) was included in this dataset, which forms a transcription complex with the intracellular domain of the amyloid precursor protein (APP) (APP_ICD) [67]. Both Notch_ICD and APP_ICD are products of *γ*-secretase cleavage. Strikingly, we also found four different subunits of the *γ*-secretase complex in our VLGR1 TAPs (see below), indicating that VLGR1 is a target for *γ*-secretase cleavage. In a parallel study, we obtained evidence that VLGR1_ICD is also cleaved and appears in the nucleus [62]. Since there is no obvious nuclear localization sequence in VLGR1 predicted by in silico analysis applying NLStradamus [68], it is probably co-shuttled with one of their binding proteins into the nucleus for fulfilling its nuclear function.

In addition, we identified several proteins that are known to participate in roles regarding pre-mRNA splicing catalyzed by the spliceosome (Figure 5d), a compositionally dynamic complex assembled stepwise on pre-mRNA [69]. We identified the RNA-binding protein Fox-1 homolog 2 (RBFOX2) as a highly reliable interactor (CRAPome value 14) which serves as an important regulator of alternative exon splicing, particularly in the nervous system [70]. In addition, we found the SNW domain containing 1 protein (SNW1), which is proposed to recruit peptidylprolyl isomerase-like 1 (PPIL1), an additional high-fidelity prey (CRAPome value 19), to the spliceosome [71,72]. To test whether VLGR1 is part of the spliceosomal core, we analyzed its interaction with the pre-mRNA-processing factors PRPF6 by pull down assays (Figure 5e). We observed the binding of HA-N-VLGR1_CTF to GFP-PRPF6 in contrast to GFP in GFP-Trap^®^ pull-downs. This exemplifies the interaction of VLGR1 with spliceosome components, as suggested by the TAP results.

Taken together, we identified VLGR1 in association with the protein networks of the functional modules of the ER, MAMs, and nucleus, suggesting multiple roles of VLGR1 in gene regulation, pre-RNA splicing, transcriptional control, and balancing Ca^2+^ homeostasis.

### 2.7. VLGR1 Is Associated with the γ-Secretase Complex

We found several subunits of the γ-secretase complex in our TAP, namely nicastrin (NCSTN), the two presenilins, PSEN1 and PSEN2, and APH1a [73], as well as the occasional subunit of the *γ*-secretase Basigin (BSG) [74]. Although the γ-secretase complex is best known for its proteolytic action on APP, resulting in amyloid β-peptide (Aβ) accumulation, it also conducts to the intramembranous proteolytic excision of numerous other single and multiple transmembrane proteins [75]. *γ*-secretase cleavage of the multi-spanning transmembrane protein polycystin 1 (PKD1) results in the release of its intracellular C-terminus, which acts as a transcriptional regulator [76,77,78,79]. There are several lines of evidence that VLGR1 is a substrate of γ-secretase, such as positively charged residues at the junction of transmembrane helix 7, which are known primary determinants of the substrate binding of the secretase, and the general presence of released VLGR1_ICD in cells [62]. In addition, the nuclear localization and prey identified in VLGR1 TAPs include various transcriptional regulators as putative targets for the ICD of VLGR1 in the nucleus [62] (Figure 5a). Therefore, it is reasonable to speculate that VLGR1 regulates transcription by a mechanism similar to the transcriptional regulation by PKD1. However, a direct experimental proof for this hypothesis is still lacking. In addition, beside *γ*-secretase itself, VLGR1 TAPs also contain validated substrates of the *γ*-secretase, such as the ephrin beta 1 (EFNB1), the insulin-like growth factor 1 receptor (IGF1R), the LDL receptor-related protein 1 (LRP1), and the amyloid precursor protein (APP). Accordingly, it would also be possible that VLGR1 is indirectly linked to the *γ*-secretase via the interaction of these receptors, or it may act as a regulatory component of the proteolytic cleavage of these substrates.

### 2.8. VLGR1 Interacts with Proteins Involved in Neurogenesis

In VLGR1 TAPs we found 161 proteins in total assigned to the GO term neurogenesis, which is in line with the high expression of VLGR1 in the developing brain [2]. These hits were reduced to 69 proteins by filtering against the CRAPome dataset (cutout ≤ 20) (Figure 6a). Fourteen of them were categorized to the GO term neuron projection guidance, which includes the Alzheimer disease-related proteins APP, three subunits of the *γ*-secretase (NCSTN, PSEN2, APH1A), and Slit guidance ligand 2 (SLIT2) (Figure 6a). SLIT2 acts as a guidance cue for retinal ganglion cells [80,81]. Interestingly, SLIT2 is related to the fibronectin leucine-rich transmembrane protein 3 (FLRT3) [82], a ligand of the ADGR Latrophilin 2 (LPHN2, ADGRL2) [83], which we also identified in VLGR1a TAPs (Appendix A). The APP-binding protein APBB1 and the protein tyrosine kinase 7 (PTK7), an inactive kinase known to participate in Wnt signaling [84], were categorized to the category positive regulators of neuron projection. In accordance with the role of VLGR1, in the outgrowth of neuron projections is our finding that the overexpression of VLGR1_CTFs and myristoyl-palmitoyl-tagged VLGR1_ICD induces the formation of long “axon-like” cell projections (Figure 6b). Nevertheless, other prey were categorized to negative regulators of neurogenesis, which includes APP, PSEN1, and the integral membrane protein 2C (ITM2C), a regulator of APP [85].

GO term categorization also showed that several prey of VLGR1 TAPs play a role in the development of the neuronal retina. These include Sidekick 2 (SDK2), which is essential for the establishment of the well-defined characteristic layers of the neuronal retina [86]. Similar to VLGR1, SDK2 harbors a C-terminal type 1 PBM responsible for binding to PDZ domains in scaffold proteins. This is also essential for SDK2 localization to synapses [87]. Evidence suggests that SDK2 and VLGR1 bind to MAGI proteins (membrane-associated guanylate kinase (MAGUK) with a reversed arrangement of protein–protein interaction domains) and the postsynaptic density protein 95 (PSD95), respectively, thereby, in this manner, clustering at the membrane of the postsynaptic terminal [87].

TAP prey also point to a role of VLGR1 in retinoid acid receptor-mediated signaling, which essential for the development of retinal photoreceptors determining the ratio of rod and cone photoreceptors in the retina [88]. We identified in VLGR1 TAPs the ligand-dependent nuclear receptor interacting factor 1 (LRIF1), which regulates the retinoic acid receptor alpha (RARA) transcription [63] and SNW1, which binds to RXR receptors, probably together with the nuclear receptor coactivator 1 (NCOA1) [89]. The interplay of VLGR1 with RAX and NCOA1 is supported by own unpublished findings from yeast-2-hybrid screens that indicated a direct interaction of VLGR1 with NCOA1. Interestingly, both SNW1 and VLGR1 have also been shown to be part of protein interactomes associated with pre-RNA splicing (see above) [90].

In summary, our data indicate that VLGR1 plays a role in neurogenesis, particularly during the development of the neuronal retina, and is implicated in axon guidance, synapse formation, and related signaling pathways.

### 2.9. VLGR1 TAPs Confirm and Specify Roles in the Function of the Mature Retina and Photoreceptor Cells

VLGR1 is highly expressed in the neuronal retina of vertebrates, mainly localized to specialized adhesion complexes such as the periciliary membrane complex at the base of the photoreceptor cilium, the retinal ribbon synapses of photoreceptor cells [5,20]. Accordingly, in VLGR1 TAPs we have identified several proteins that are attributed to the *Cellular Component* GO terms, the *photoreceptor outer segment, axoneme, photoreceptor connecting cilium, photoreceptor inner segment,* and *neuronal postsynaptic density* (Figure 7a,b). Among the enriched GO terms in the *Biological Process* category, we found several terms connected to retinal or ciliary function. These are, for example, *cilium organization, visual perception, retina development in the camera-type eye, retina homeostasis, phototransduction, neurotransmitter secretion,* and the *regulation of the synapse structure or activity* (Figure 7c). Data analysis with the cytoscape plugin *stringApp* (http://apps.cytoscape.org/apps/stringapp, accessed on 10 September 2017) demonstrated that the interconnection of about half of these proteins is supported by STRING data (https://string-db.org/, accessed on 10 September 2017). Notably, TAP prey assigned with GO terms related to the *photoreceptor outer segment, connecting cilium, cilium organization,* and *visual perception* were exclusively found in the dataset of the full-length protein VLGR1a. Prey assigned to the *inner segment, phototransduction,* and *retinal development* were identified in all three datasets, the full-length VLGR1a and the two VLGR1_CTFs, whereas prey assigned to the *neuronal post-synaptic density* and *regulation of the synaptic structure or activity* were mainly found with the VLGR1_CTFs.

These data indicate an important role of VLGR1 in retinal photoreceptor cell function. This is further strengthened by the fact that mutations in the VLGR1 gene cause USH2C, which is also characterized by a strong ocular component leading to vision loss [4]. As already discussed above, VLGR1 TAPs indicate the interaction with proteins associated with syndromic and non-syndromic retinal ciliopathies, such as KIF11 [91], Nephrocystin 5 (IQCB1/NPHP5), associated with Senior-Løken syndrome [92], and Retinitis pigmentosa 1 (RP1), the product of the autosomal dominant Retinitis pigmentosa *RP1* gene [93,94]. Furthermore, VLGR1 TAPs contain Wolframin (WFS1) which is highly expressed in the human retina [95] and it is, similar to VLGR1, part of the photoreceptor ciliome [96]. In addition, as in VLGR1, mutations in *WFS1* can cause deafness [97].

### 2.10. TAP Data Support a Role of VLGR1 Associated with Epilepsy

VLGR1 is highly expressed in the central nervous system [2]. VLGR1 mutant mice are prone to audiogenic seizures and so considered as models for epilepsy [11,12,13]. In addition, there is increasing evidence that the haploinsufficiency of VLGR1 and epilepsy are related in humans [7,9,10,98]. This prompted us to screen our TAP data for candidate molecules related to the different forms of epilepsy. Previous studies indicated that defects in the *CHRNA4* gene, which encodes a subunit of the high-affinity nicotinic acetylcholine receptor (nAChR), lead to autosomal dominant nocturnal frontal lobe epilepsy (ADNFLE) [99]. Interestingly, in our VLGR1_CTF TAP data, CHRNA5, another subunit of the nAChR, which has not been associated with ADNFLE so far, was present. However, in these VLGR1_CTF TAPs, we additionally found the voltage-gated Cl- channels, CLCN1, CLCN3, and CLCN7. Missense variants in *CLCN7* are enriched in epilepsy patients, making them candidates for the disease [100]. Furthermore, we found NIPA2 (non-imprinted in the Prader-Willi/Angelman syndrome region protein 2) in all VLGR1_CTF TAPs. NIPA2 is a highly selective Mg^2+^ transporter and *NIPA2* mutations were previously described within a population of patients with childhood absence epilepsy [101]. Taken together, our findings are in line with a role of VLGR1 in the pathogenesis in epilepsy.

### 2.11. VLGR1 Protein Networks Are Linked to Neuron Death and Degenerative Diseases of the Central Nervous System

In line with a role in neuronal degeneration, TAP prey were assigned to the GO term *neuron death* (Figure 8). After filtering against the CRAPome (cutout ≤ 20) 24 proteins remained that belong to this category. These include, e.g., 24-Dehydrocholesterol reductase (DHCR24), Sigma non-opioid intracellular receptor 1 (SIGMAR1), and Wolframin (WFS1), which have cell protective functions. Mutations in these proteins are associated with severe diseases, such as Alzheimer’s disease, Amyotrophic lateral sclerosis (ALS), and Wolfram syndrome [102,103,104].

### 2.12. VLGR1 Is Found in Alzheimer’s Disease-Related Protein Complexes

In VLGR1 TAPs, we identified several molecules in whose genes mutations cause early-onset familial Alzheimer’s disease (AD). An amyloid precursor protein (APP) occurred as a prey in one of our TAPs with SF-N-VLGR1_CTF. Mutations in the APP are related to AD, whereby APP processing by secretases plays a critical role in the Alzheimer’s disease’s progression [105]. In addition to the γ-secretase discussed above, we also found additional proteins that are involved in APP processing, regulation, and/or signaling. We identified, for example, the metalloproteases ADAM10 and ADAM17 that both cleave APP in the extracellular domain, preceding γ-secretase cleavage in the non-amyloidogenic pathway [106,107]. Another regulator of APP found in the VLGR1 TAPs is LRP1 which is involved in endocytosis of the released APP770 fragment, and its protein expression is decreased in Alzheimer’s patients [108]. Furthermore, we found the amyloid beta precursor protein-binding family B member 1 (APBB1) that binds to the intracellular fragment of APP (APP_ICD). Both LRP1 and APBB1 act together in gene expression [109,110]. APBB1 also co-localizes with APP in the ER and/or Golgi and, presumably, in endosomes, where it regulates the APP metabolism [111]. Interestingly, the above-mentioned TAP prey and γ-secretase substrate LRP1 modulates APBB1/APP-mediated gene activation [112,113].

The TAP prey DHCR24 (24-Dehydrocholesterol reductase) is also related to Alezheimer’s disease and plays a role in the cholesterol metabolic pathway [114]. It has a protective effect against Aβ-induced toxicity and is downregulated in Alzheimer’s patients [108]. Last but not least, we found the protein four and a half LIM domains 2 (FHL2), which binds to both ADAM17 [115] and PSEN2 [116], and thereby could indirectly regulate APP cleavage.

In summary, we identified a whole set of VLGR1 interactors that are related to APP metabolism. Whether VLGR1 is itself processed by γ-secretase and/or metalloproteases, or whether it is implicated in APP-mediated pathways in a different way, remains to be elucidated [102,103,104].

## 3. Materials and Methods

### 3.1. Plasmids

Plasmids used for tandem affinity purifications were coded for Strep II-FLAG (SF)-tagged human VLGR1a (Uni-Prot ID Q8WXG9-2, aa 1- 1967), VLGR1_CTF (Uniprot ID Q8WXG9-1, aa 5891-6306), and VLGR1_ICD (Uniprot ID Q8WXG9-1, aa 6155-6306). The SF-tag was inserted between aa 66 and 67 in the VLGR1a construct, N-terminally and C-terminally fused in VLGR1_CTF, and N-terminally fused in VLGR1_ICD. For myristoyl-palmitoyl-tagged VLGR1_ICD, VLGR1_ICD was subcloned into the MyrPalm-eCFP vector (plasmid 14867, Addgene, Cambridge, MA, USA). For 3xHA-tagged VLGR1 constructs, VLGR1_CTF and VLGR1_ICD were subcloned into the p3xHA/DEST vector. Receptors for second messenger assays were sublconed into pcDps vector and HA- and Flag-tagged at the N- and C-terminus, respectively. Truncated CTF receptor version was fused to the N-terminus of P2Y12 to ensure proper membrane expression, as described in Liebscher et al. [29].

### 3.2. Cell Culture

HEK293T, HeLa, and hTERT-RPE1 cells were cultured in Dulbecco’s modified Eagle’s medium (DMEM and DMEM-F12, respectively) containing 10% heat-inactivated fetal calf serum (FCS). Primary astrocytes were isolated from mouse brains and cultured, as described in [117]. Cells were transfected with GeneJuice^®^ (Merck Millipore, Darmstadt, Germany) according to manufacturer’s instructions, if not indicated differently.

### 3.3. Tandem Affinity Purification (TAP)

The tandem affinity purification (TAP) was performed as described in [118]. Four TAPs were performed for VLGR1a, three TAPs for each VLGR1_CTF construct and two TAPs for VLGR1_ICD. In brief, SF-tagged proteins were overexpressed in HEK293T cells for 48 h. Mock treated cells were used as a control. The cells were lysed and the lysate was cleared by centrifugation. The supernatant was then subjected to a two-step purification on Strep-Tactin^®^Superflow^®^ beads (IBA, Göttingen, Germany) and anti-FLAG M2 agarose beads (Sigma-Aldrich, Hamburg, Germany). Competitive elution was achieved by desbiothin (IBA, Göttingen, Germany) in the first step and FLAG^®^ peptide (Sigma-Aldrich, Hamburg, Germany) in the second step. The eluate was precipitated by methanol–chloroform and then subjected to mass spectrometric analysis.

### 3.4. Mass Spectrometry

Mass spectrometry was performed as previously described by [38]. In brief, SF-TAP-purified protein complexes were solubilized before subjecting to trypsin cleavage. Resulting peptides were desalted and purified using stage tips before separation on a Dionex RSLC system. Eluted peptides were directly ionized by Nano spray ionization and detected by a LTQ Orbitrap Velos mass spectrometer (Thermo Fisher Scientific, Waltham, MA, USA). We searched the raw spectra against the human SwissProt database using Mascot, and verified the results by Scaffold (version Scaffold 4.02.01, Proteome Software Inc., Portland, OR, USA) to validate MS/MS-based peptide and protein identifications.

### 3.5. Data Processing

Mass spectrometry data of the different VLGR1 fragments were compared to the according data for mock-transfected cells. Proteins that occurred in the mock dataset were not considered for subsequent analysis. Datasets of identical VLGR1 fragments were combined for further analysis. Gene names (according to HGNC) of VLGR1 prey were used as input for the Cytoscape plugins, STRING and ClueGO. The parameter confidence (score) cutout was set to 0.40 and the parameter maximum number of interactors was set to 0 for STRING analysis. ClueGO v2.3.3 was used for Gene Ontology (GO) term enrichment analysis. Network specificity was set to default (medium).

### 3.6. Antibodies

The following antibodies were used: rabbit anti-VLGR1 [20], mouse anti-Vinculin (Merck KGaA, Darmstadt, Germany, clone hVIN-1), mouse anti-CLIMP63 (Enzo Life Sciences, Lörrach, Germany, G1/296), rat anti-HA (Roche, clone 3F10), mouse anti-FLAG M2 (Merck KGaA, clone M2), rabbit anti-GFP (kind gift from Dr. Clay Smith, University of Florida), mouse anti-α-Tubulin (Abcam, Cambridge, UK, clone DM1A), and goat anti-Pericentrin 2 (Santa Cruz Biotechnology, Dallas, TX, USA, clone C-16). Secondary antibodies conjugated to Alexa 488, Alexa 568, or Alexa 647 were purchased from Molecular Probes (Life Technologies, Darmstadt, Germany) or from Rockland Inc. (Gilbertsville, PA, USA). Nuclear DNA was stained with DAPI (1 mg/mL) (Merck KGaA).

### 3.7. Immunocytochemistry

Cells were fixed and permeabilized in ice cold methanol for 10 min and washed with PBS. After washing, the cells were covered with blocking solution and incubated overnight with the primary antibody at 4 °C. Cells were washed and then incubated with the secondary antibody in blocking solution containing DAPI for 1.5 h at room temperature. After washing, sections were mounted in Mowiol (Roth). Specimens were analyzed on a Leica DM6000B microscope, and images were processed with Leica imaging software and Adobe Photoshop CS (intensity adjustment).

### 3.8. siRNA Knock Down

For VLGR1 knock-down, hTERT-RPE1 cells (1 × 105) were seeded on 6-well plates. After 18 h, cells were transfected with 100 µmol of a siRNA pool targeted against VLGR1 (L-005656-00-0005, Dharmacon, Lafayette, CO, USA) and non-targeting siRNA (D-001810-10-05), respectively, using Lipofectamin^®^ RNAiMAX (Life Technologies, Darmstadt, Germany), according to the manufacturer’s protocol. Concomitant with siRNA knock-down, cells were starved for 72 h in OptiMEM^®^ to induce cilia formation.

### 3.9. GFP-Trap^®^

GFP-fused proteins were immobilized on nanobody GFP-Trap^®^ agarose beads (ChromoTek, Planegg-Martinsried, Germany) and used for co-precipitation assays according to the manufacturer’s protocol. Briefly, cell lysates from co-transfected HEK293T cells (GFP-tagged proteins with HA-tagged proteins) were suspended in lysis buffer, spun down, and the supernatant was diluted to 1 mL dilution buffer. Next, 50 µm were separated as input (total cell lysate) and samples were added to equilibrated beads for 2 h at 4 °C under constant shaking. After washing, precipitated protein complexes were eluted with SDS-sample buffer and subjected to SDS–PAGE and Western blotting.

### 3.10. Immunoprecipitation

For co-IP, HA-PRPF31 was co-expressed with SF-N-VLGR1_ICD in HEK293T cells and lysed in Triton-X-100 lysis buffer containing PI-mix. Co-IP was performed using anti-HA affinity gel (Biotool, Munich, Germany), according to the manufacturer’s protocol. Briefly, cell lysates were incubated with equilibrated HA-affinity beads for 2 h at 4 °C. After three washing steps, samples were eluted with SDS-sample buffer and subjected to SDS-page and Western blot analysis.

### 3.11. Second Messenger Assays

Measurements of cAMP and IP were performed as described previously [119]. In brief, HEK293T cells were split into 12-well plates (IP assay) and 48-well plates (cAMP assay) and transfected with LipofectamineTM2000 (Invitrogen, Paisley, UK), according to the manufacturer’s protocol. For the IP assay, cells were incubated with 2 μCi/mL myo-[3H]inositol (18.6 Ci/mmol, PerkinElmer Life Sciences, Waltham, MA, USA) for 24 h. After washing with 10 mM LiCl in serum-free DMEM, cells were incubated with 10 mM LiCl for 1 h at 37 °C. Intracellular IP levels were determined by anion-exchange chromatography, as described in [120]. IP accumulation data were analyzed using GraphPad Prism version 4.0 for Windows (GraphPad Software, San Diego, CA, USA). For the cAMP assay, transfected cells were incubated with 3-isobutyl-methyl-xanthine (1 mM)-containing medium for 1 h. Cells were lysed in LI buffer (PerkinElmer Life Sciences, Monza, Italy) and frozen at −20 °C until measurement. The cAMP concentration was determined with the Alpha Screen cAMP assay kit (PerkinElmer Life Sciences), according to the manufacturer’s protocol, and measured with the Fusion AlphaScreen multilabel reader (PerkinElmer Life Sciences). The surface expression of the HA-tagged receptors was determined by an indirect cellular enzyme-linked immunosorbent assay (ELISA), as described in [121]. Peptide synthesis was performed, as described in [122].

## 4. Conclusions

Our results, obtained by systematic affinity proteomics, point to diverse roles of VLGR1, depending on its cellular localization and structural or state of activation (Figure 9). We provide evidence that VLGR1 harbors a tethered agonist sequence, which mediates a switch in Gα-mediated signaling. Our data also suggest that VLGR1 is active in functional modules of the cell that include cellular Ca^2+^ homeostasis, transcriptional regulation, pre-mRNA splicing, cell differentiation, and primary cilia integrity. In addition, our study provides evidence for the role of VLGR1 in the maintenance of sensory-neuronal systems, and the identified interactions with disease molecules support previously found associations with diseases such as Usher syndrome and epilepsy, but also suggest links to other neuronal disorders, such as Alzheimer’s disease. Whether these functions are mediated via the activation of signaling cascades, the adhesive function of VLGR1, or a combination of both will need to be elucidated in future detailed studies.

## Figures and Tables

**Figure 1 molecules-27-03108-f001:**
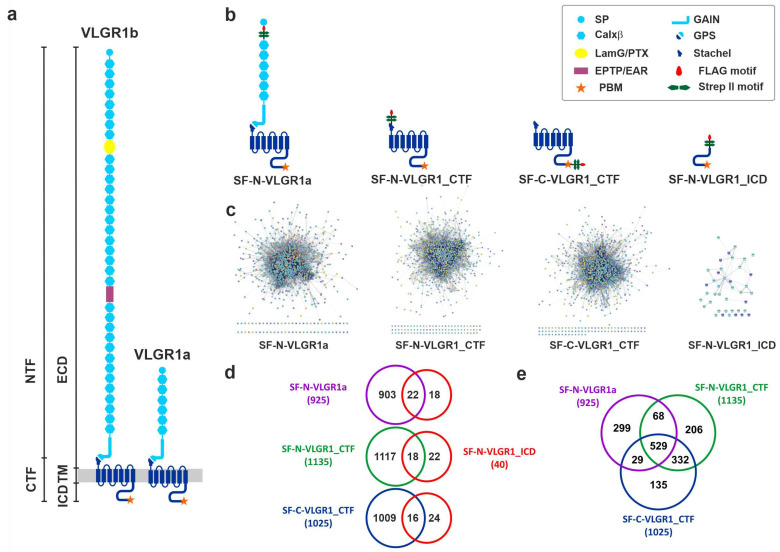
**VLGR1 domain structures and VLGR1-associated protein networks.** (**a**) The two long isoforms of VLGR1, the full-length protein VLGR1b, and the shorter isoform VLGR1a can be autocleaved at the GPS (G-protein-coupled receptor proteolytic site) into the following fragments: N-terminal fragment (NTF) and the C-terminal fragment (CTF), which are composed of a short intracellular domain (ICD), the seven transmembrane domain (TM), and the long extracellular domain (ECD). (**b**) N- or C-terminally Strep II- FLAG (SF)-tagged VLGR1 constructs were used as baits. (**c**) Protein networks of prey identified with SF-N-VLGR1a (863 out of 925 prey proteins are interconnected, based on the STRING database (https://string-db.org/, accessed on 10 September 2017), SF-N-VLGR1_CTF (1054 out of 1135 prey proteins are interconnected), SF-C-VLGR1_CTF (939 of 1025 prey proteins are interconnected), and SF-N-VLGR1_ICD (29 out of 40 prey proteins are interconnected). (**d**,**e**) Venn diagrams of VLGR1 prey revealing overlaps between the interactomes found for the VLGR1 constructs.

**Figure 2 molecules-27-03108-f002:**
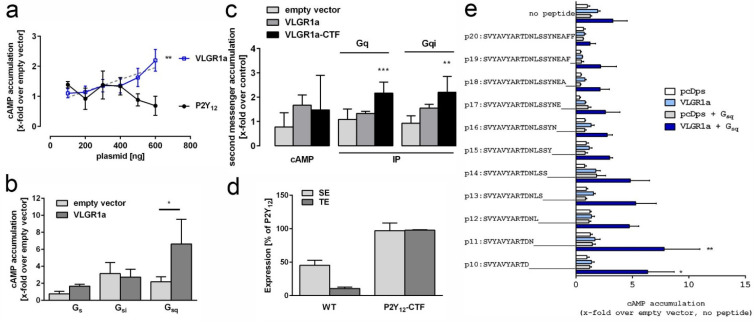
**Signal transduction and activation of VLGR1.** (**a**) HEK293T cells were transfected with increasing amounts (100–600 ng/well) of plasmids encoding either human VLGR1 or human P2Y12. VLGR1, but not P2Y12, a known Gi coupling receptor, caused a dose-dependent increase of cAMP. Statistics displayed significant linear regression for VLGR1. (**b**) Coupling analysis of VLGR1a using chimeric G proteins in cAMP accumulation assay revealed significant 2nd messenger accumulation with a chimera that couples to a Gq-binding receptor. (**c**) Comparison of basal 2nd messenger production of full length and CTF, where the NTF was replaced with the N-terminus of P2Y12 to ensure proper surface expression of the mutant. Constitutive activity of the CTF is observed in IP accumulation assay with and without a Gqi chimeric protein, indicating coupling to Gq and Gi. (**d**) Cell surface (SE) and total cell expression (TE) of full length and CTF mutant constructs of VLGR1 in relation to P2Y12 (100%), which served as positive control. (**e**) Screening for VLGR1 *Stachel*-mimicking peptides. A *Stachel*-derived peptide library of VLGR1 was tested in cAMP with and without the addition of a Gsq chimeric G protein. Peptides 10 and 11 amino acids long were found to significantly activate VLGR1 in comparison to pcDps when the Gsq chimera was added. (**a**–**e**): Data are given as means ± S.D. of three independent experiments each performed in triplicate. (**b**,**c**): Statistics was performed using one-way ANOVA with Bonferroni as post-hoc test and (**d**) utilized one-way ANOVA with Sidak’s multiple comparison test. p-values in b, c and e: * *p* ≤ 0.05, ** *p* ≤ 0.01, *** *p* ≤ 0.001.

**Figure 3 molecules-27-03108-f003:**
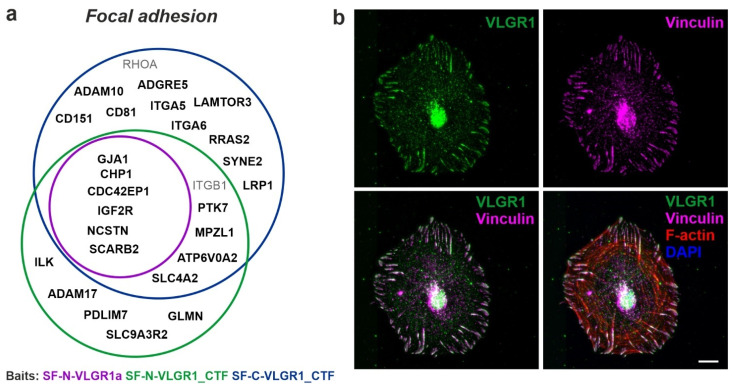
**Association of VLGR1 with focal adhesions.** (**a**) Venn diagram of VLGR1 prey assigned to the GO term *focal adhesion* in the category *Cellular Component*. Black font: CRAPome value ≤20; Grey font: CRAPome value > 20. (**b**) VLGR1 is expressed at focal adhesions (FA) in primary astrocytes derived from murine brain, where it co-localizes with the FA marker Vinculin and actin stress fibers. Scale bar: 10 µm.

**Figure 4 molecules-27-03108-f004:**
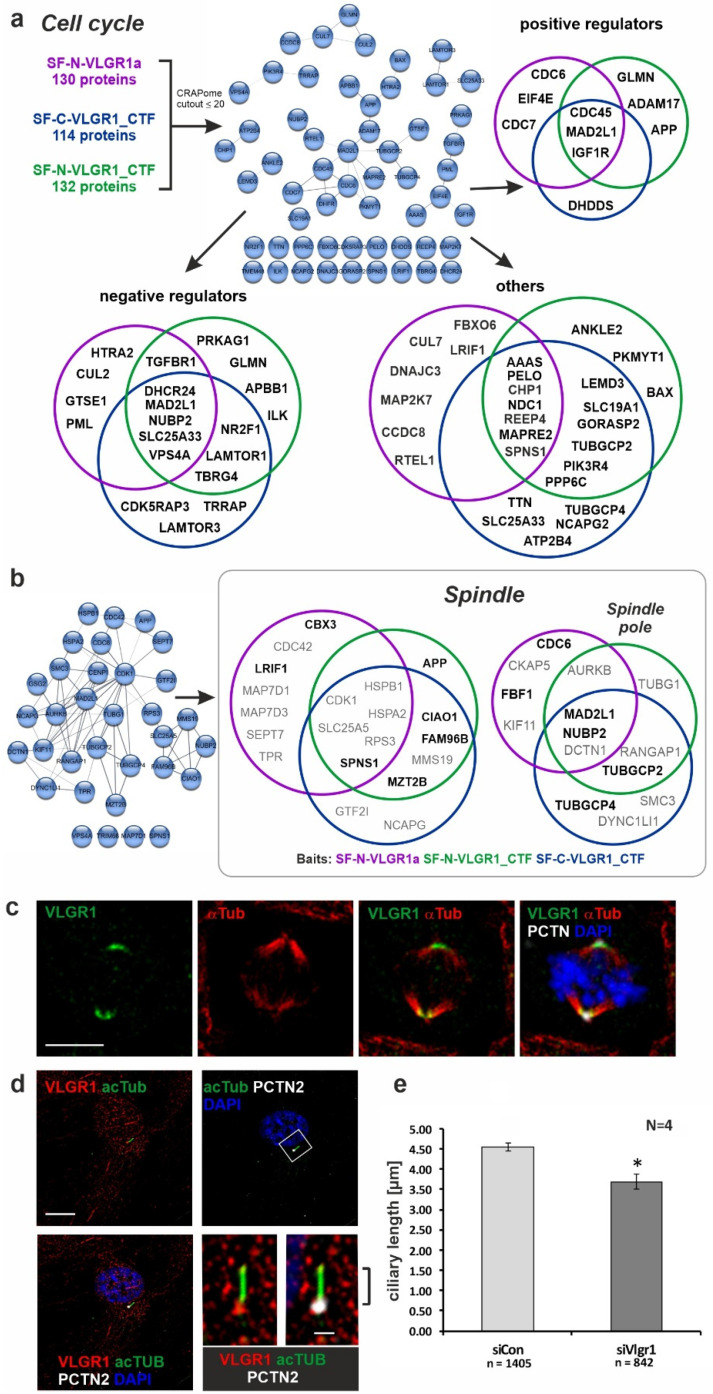
**Association of VLGR1 with cell cycle regulators, mitotic spindle components, and primary cilia.** (**a**) Venn diagrams of VLGR1 prey involved in cell cycle regulation. Interactions between these regulators are visualized in a STRING (https://string-db.org/, accessed on 10 September 2017) network. (**b**) VLGR1 prey that are assigned to the GO terms spindle and spindle pole in the category Cellular Component. Black font: CRAPome value ≤ 20; Grey font: CRAPome value > 20. (**c**) Triple immunofluorescence staining of VLGR1 (green), the spindle microtubules by α-tubulin (α-Tub, red), and the centriole/spindle pole marker protein pericentrin-2 (PCTN, white) in RPE1 cells reveals the localization of VLGR1 at spindle poles. Chromosomal DNA is stained by DAPI (blue). (**d**) Triple immunofluorescence staining revealed VLGR1 (red) co-localization with PCTN2 at the base of primary cilia in RPE1 cells. (**e**) siRNA-mediated knock-down of VLGR1 in RPE1 cells results in decrease of the ciliary length. Scale bars: 10 µm and 2 µm (magnified primary cilium). siCon, control siRNA (non-targeting); siVlgr1, siRNA directed against VLGR1; * *p* < 0.05.

**Figure 5 molecules-27-03108-f005:**
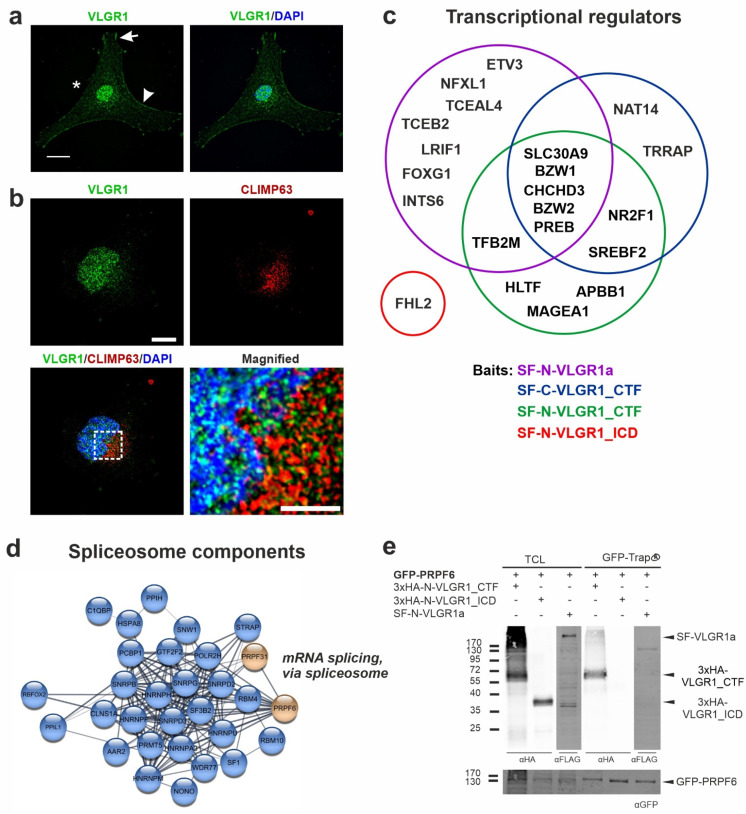
**VLGR1 nuclear localization and its interaction with transcriptional regulators and components of the spliceosome.** (**a**) Indirect immunofluorescence of VLGR1 counter stained for the nucleus by DAPI (blue) in astrocytes derived from mouse brains. VLGR1 (green) is localized in the plasma membrane (*arrowhead*) in the cytoplasm, in focal adhesions (*arrow*), and in the nucleoplasm of the nucleus (*asterisk*). (**b**) Double immunofluorescence of VLGR1 (green) and the ER marker CLIMP63 (red), counter stained for nuclear DNA by DAPI (blue) in HeLa cells demonstrating prominent VLGR1 localization in the ER. (**c**) Venn diagram of VLGR1 prey with a function in transcriptional regulation (CRAPome value ≤ 20). (**d**) STRING network of VLGR1 TAP prey assigned to the GO term *mRNA splicing,* via *spliceosome*. (**e**) Validation of the interaction of VLGR1 with the spliceosome component PRPF6. GFP-PRPF6 pulled down VLGR1_CTF in a GFP-Trap^®^ assay. TCL: total cell lysate, CTF: C-terminal fragment, ICD; intracellular domain, SF: Strep II-FLAG. Scale bars in (**a**,**b**): 10 µm.

**Figure 6 molecules-27-03108-f006:**
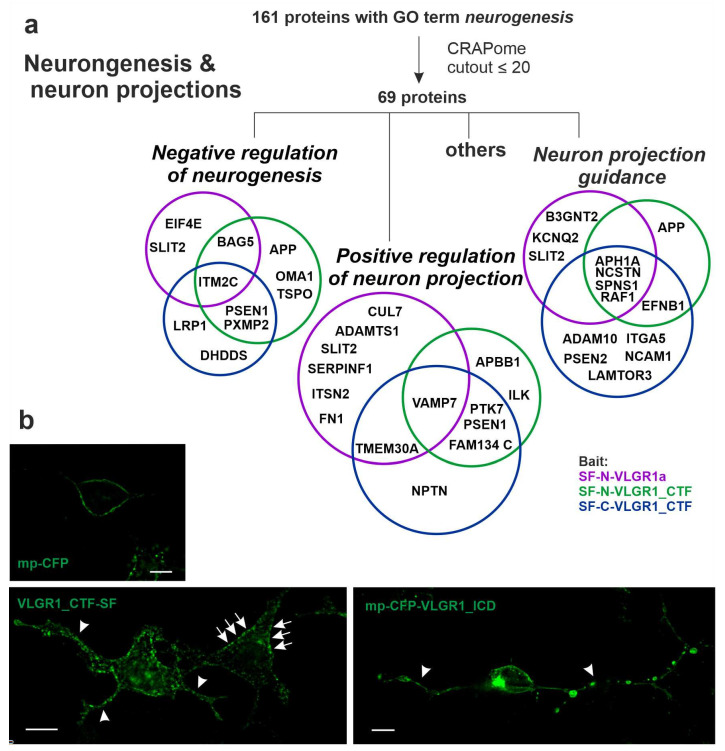
**VLGR1 associates with neurogenesis.** (**a**) Venn diagrams of VLGR1 prey that are assigned to the GO term neurogenesis in the category Biological Process. (**b**) Overexpression of VLGR1_CTF-SF and myristoyl-palmitoyl-(mp) VLGR1_ICD induces “axon-like outgrowth” (arrowheads) which are not found in myristoyl-palmitoyl-CFP transfected HEK293T cells. Note that VLGR1_CTF-SF is also localized at the plasma membrane (*arrows*). Scale bars: 10 µm.

**Figure 7 molecules-27-03108-f007:**
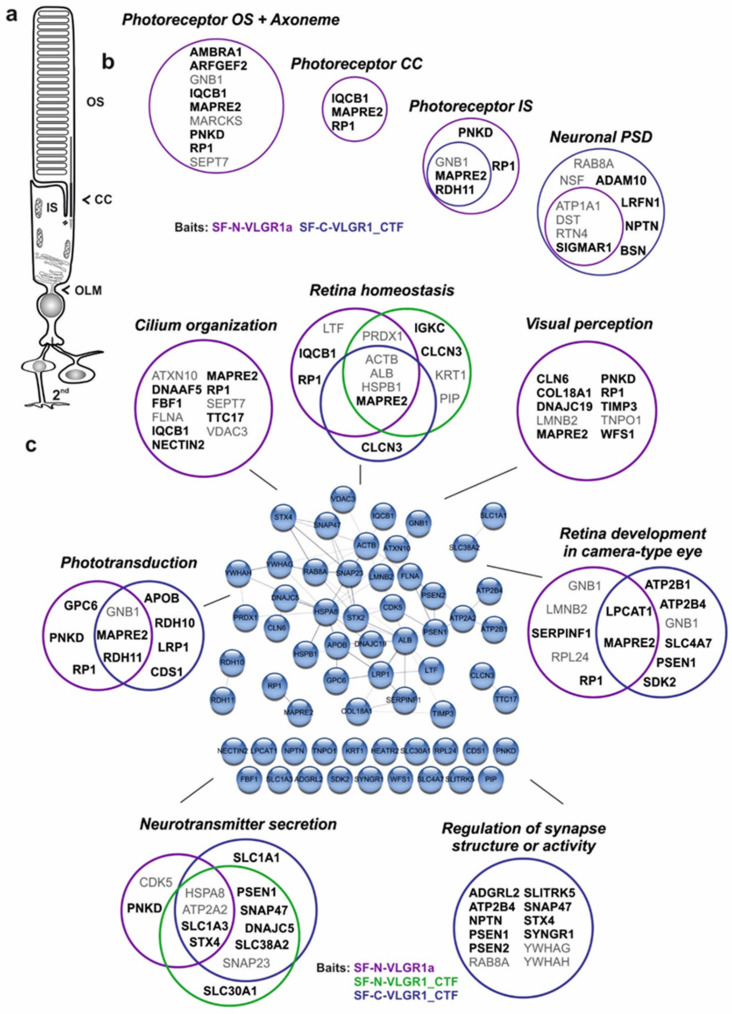
**VLGR1 prey associates with molecules related to photoreceptors and retinal functions.** (**a**) Scheme of a rod photoreceptor cell, (**b**) VLGR1 prey assigned with GO terms in the Cellular Component category that are related to photoreceptor cells and cell synapses. OS, outer segment; CC, connecting cilium; IS, inner segment; OLM, outer limiting membrane; PSD, postsynaptic density; Ax, axoneme. Black font: CRAPome value ≤ 20; Grey font: CRAPome value > 20. (**c**) Relation of VLGR1 prey to retinal function. Venn diagrams of VLGR1 prey that are assigned to retina-related GO terms in the Biological Process category. Black font: CRAPome value ≤ 20; Grey font: CRAPome value > 20.

**Figure 8 molecules-27-03108-f008:**
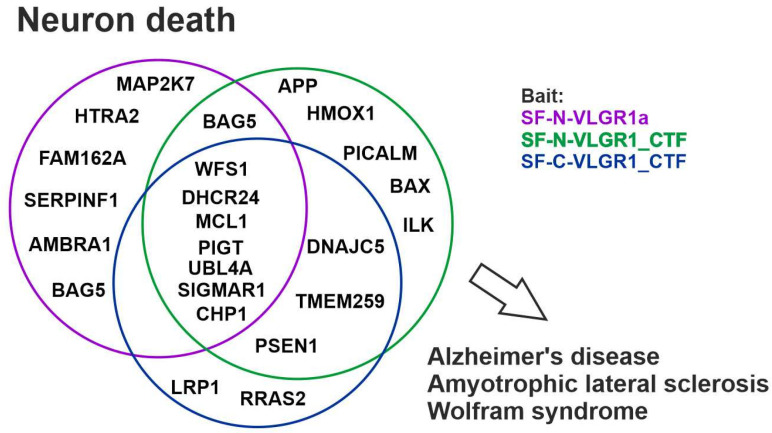
**VLGR1 associates with neuron death.** VLGR1 prey are associated with neuron integrity. Venn diagram of VLGR1 prey assigned with the GO term *neuron death*. Some of these proteins are related to neurodegenerative diseases such as Alzheimer’s disease, Amyotrophic lateral sclerosis, and Wolfram syndrome.

**Figure 9 molecules-27-03108-f009:**
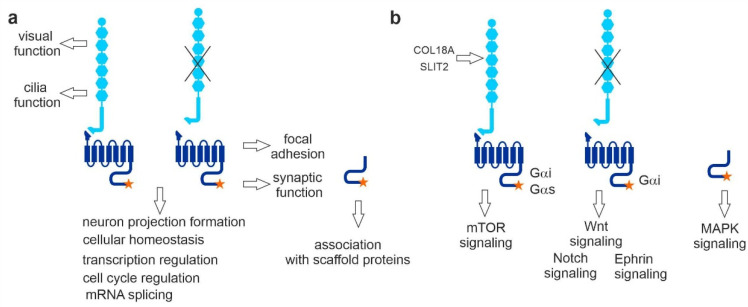
Summary and conclusions from the present affinity capture approach to VLGR1 function and for VLGR1 signaling pathways. (**a**) TAP prey related to ciliary and retinal function were mainly found with full-length VLGR1a. VLGR1_CTF was sufficient for focal adhesion and synaptic protein binding. Both full-length and VLGR1_CTF interact with proteins involved in neuron projection formation, cellular homeostasis, transcription regulation, cell cycle regulation, and pre-mRNA splicing. VLGR1_ICD binds to PDZ domain-containing scaffold proteins. (**b**) COL18A1 and SLIT2 are potential extracellular ligands for VLGR1, interacting with its NTF. VLGR1 full-length was linked to HIF-1 signaling coupled to Gαi and Gαs. VLGR1_CTF is coupled to Gαi and linked to Wnt, Notch, and Ephrin signaling; VLGR1_ICD seems to be linked to MAPK signaling.

**Table 1 molecules-27-03108-t001:** **VLGR1 is connected to several cellular signaling pathways** (selection is based on KEGG pathway and Biological Process enrichment analysis, see also Appendix A). Prey proteins with CRAPome values ≤ 20 are shown. Baits: SF-N-VLGR1a (full-length VLGR1a), SF-N-VLGR1_CTF (N-CTF), and SF-C-VLGR1_CTF (C-CTF).

Pathways	Prey Proteins	VLGR1a	Baits N-CTF	C-CTF	Prey Proteins	VLGR1a	Baits N-CTF	C-CTF
**AMPK pathway:** *coordinates cell growth, * *autophagy and metabolism*	CPT1A G6PC3 HMGCR IGF1R PCK2	+ + +	+ + + +	+ + + +	PIK3R2 PRKAG1 SCD SCD5	+ + +	+ +	+ +
**Ephrin signaling:** *development*	ADAM10 APH1A EFNB1	+	+ +	+ + +	NCSTN PSEN2	+	+	+ +
**FoxO pathway:** *regulation of cell cycle, apoptosis and metabolism*	FOXG1 G6PC3 IGF1R PIK3R2	+ + +	+ +		PCK2 PRKAG1 RAF1 TGFBR1	+ + +	+ + +	
**HIF 1 pathway:** *regulates hypoxia inducible genes under hypoxic conditions*	CUL2 EIF4E HK2 HMOX1	+ + +	+ +	+	IGF1R PIK3R2 TCEB1 TCEB2	+ + + +	+	+
**Insulin receptor pathway:** *controls critical energy functions such as glucose and lipid metabolism, connected to FoxO signaling pathway*	ATP6AP1 ATP6V0A1 ATP6V0A2 ATP6V0D1 ATP6V1F ATP6V1H EIF4E IGF1R LAMTOR1	+ + +	+ + + + + + + +	+ + + + + + + +	LAMTOR3 NCAM1 PIGU PIK3R2 PIK3R4 PRKAG1 RAF1 SPNS1	+ + +	+ + + + +	+ + + + + +
**Integrin signaling:** *development, tissue maintenance and repair*	ADAM10 ADAMTS CD47	+		+ +	ITGA5 ITGA6 SPNS1	+	+	+ +
**MAPK pathway:** *involved in proliferation, differentiation, motility, stress response,* *apoptosis and survival*	ARL6IP5 ATP6AP1 ATP6AP2 CD81 CDK5RAP3 FN1 FZD7 GDF15 HMGCR IGF1R ILK INHBE IRAK1 LAMTOR1 LAMTOR3	+ + + + + +	+ + + + + + + + +	+ + + + + + + + +	LEMD2 MAP2K7 MTCH1 NCAM1 NPTN PIGU PPM1L PSEN1 RAF1 RHBDD3 SPNS1 SYNJ2BP TAB2 TGFBR1 VRK2	+ + + + + + +	+ + + + + + + + + + +	+ + + + + + + + + +
**mTOR pathway:** *senses and integrates nutritional and environmental cues*	EIF4E LAMTOR1 LAMTOR3P	+	+	+ +	K3R2 RICTOR	+ +		
**Neurotrophin pathway:** *regulates survival, development and function of neurons*	ADAM17 APH1A IRAK1 LAMTOR3 NCAM1 NCSTN PIGU	+ + +	+ + + + +	+ + + + + +	PIK3R2 PSEN2 RAF1 RICTOR SMPD2 SORT1 SPNS1	+ + + + +	+ + + +	+ + + +
**Notch pathway:** *regulates cell fate determination during development and maintains adult tissue homeostasis*	ADAM10 ADAM17 APH1A APP GALNT11 GOLT1B MAGEA1	+	+ + + + +	+ + + +	NCSTN PSEN1 PSEN2 SEL1L SYNJ2BP TSPAN15	+	+ + + + +	+ + + + + +
**SMAD pathway:** *regulates cell growth, differentiation & development*	GDF15 INHBE LNPEP	+ +	+	+	NCEH1 SLC33A1 STOML1	+	+ +	+ +
**Sphingolipid pathway:** *regulate cellular responses to stress*	ABCC1 BAX CERS1 CERS2 CERS4 CERS5 CERS6 DEGS1 PIK3R2	+ + +	+ + + + + + +	+ + + + + + +	PLCB4 PLD2 RAF1 SGPL1 SGPP1 SMPD2 SPTLC1 SPTLC2	+ + + +	+ + + + + + +	+ + + + + +
**Wnt pathway:** *regulates cell migration, cell polarity, neural patterning and organogenesis*	ATP6AP2 FZD7 GPC6 ILK LRP1 MESDC2	+ +	+ + +	+ +	PSEN1 PTK7 SMO SPNS1 UBR5 WLS	+ +	+ + + + +	+ + + + +

## Data Availability

All data obtained in the present work are included in the main body of the publication or are provided as Appendix A.

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
