# Peer review of "Affinity Proteomics Identifies Interaction Partners and Defines Novel Insights into the Function of the Adhesion GPCR VLGR1/ADGRV1"

_molecules, 2022, doi:10.3390/molecules27103108_

Round 1

Reviewer 1 Report

VLGR1 is the member of the adhesion G protein-coupled receptor family. aGPCRs have been linked to a wide variety of different diseases and regulate many important physiological processes throughout the organism. Despite many years of research, the molecular mechanisms of aGPCRs functioning remain elusive. Adhesion GPCRs are part of complex networks of interacting proteins. Therefore, understanding the signaling pathways and searching for the molecular partners of aGPCRs has crucial role for achieving a fundamental understanding of receptor function and future drug discovery. The authors in their work “Affinity proteomics identifies interaction partners and defines novel insights into the function of the adhesion GPCR VLGR1/ADGRV1” provided affinity proteomics to identify potential binding partners and ligands of VLGR1. Despite the relevance of the chosen approach, the results has serious issues to be addressed.

Major comments:

  1. The main issue is the selection of the HEK293 cells for the expression and subsequent purification of VLGR1 and its truncated forms in complexes with interacting proteins. Though in their previous work Authors used hTERT-immortalized retinal pigment epithelial cells (hTERT-RPE1) (Kusuluri, D.K.; Guler, B.E.; Knapp, B.; Horn, N.; Boldt, K.; Ueffing, M.; Aust, G.; Wolfrum, U. Adhesion G protein-coupled 839 receptor VLGR1/ADGRV1 regulates cell spreading and migration by mechanosensing at focal adhesions. iScience 2021, 840 24, 102283, doi:10.1016/j.isci.2021.102283). HEK293 cells are epithelial kidney cancer cells. These cells are not specific for VLGR1 expression. The results obtained cannot be interpreted unambiguously. Thus, RP1 and NR2F1 proteins specifically expressed in photoreceptor cells were identified as interacting partners of VRGL1 in HEK293.
  2. The discussion of the most obtained data looks rather speculative in the absence of proper experimental data. It concerns conclusions about the role of VLGR1 in Alzheimer’s disease, epilepsy, and in function of the mature retina and photoreceptor cells.
  3. Control of proper membrane expression of VLGR1 transmembrane forms is absent. It is not clear what was used as a negative control for affinity purification (mock-transfected cells?).
  4. Previous work of the Authors mentioned above (Kusuluri, D.K. et al, iScience 2021, 840 24, 102283, doi:10.1016/j.isci.2021.102283) was based on the same TAP approach. Were the G-proteins and USH identified as interacting proteins in RPE1 cells?
  5. Materials and methods are not described in sufficient detail. Description of tandem affinity purification (TAP) and mass spectrometry is completely
  6. Figure legends are insufficient informative.
  7. Fig 3b is almost a duplicate of a figure from previous publication (Kusuluri, D.K. et al, iScience 2021, 840 24, 102283, doi:10.1016/j.isci.2021.102283) and does not contain any additional information.
  8. The cells in Fig 5a referred as HEK293 in the text body and as astrocytes derived from mouse brains in the figure legend. Tracers for cell compartment labeling should be used to confirm protein localization.
  9. The abbreviations are absent in Fig 5d, e legend. There are no protein bands corresponding to the proteins SF-N-VLGR1a, SF-N-VLGR1-CTF in total cell lysates. The blot even does not contain area of 180 kDa, where the extracellular part of SF-N-VLGR1a should be observed. Thus the question arises whether the anti-HA immunoprecipitation was performed correctly.

Reviewer 2 Report

This comprehensive article describes data the adhesion GPCR, VLGR1/ADGRV1. It provides leads for further investigation of the functions of these interactions.

I have a few suggestions:

  1. For GPCR forward trafficking, molecular chaperones are critical (see PMID 29442594). Can the authors discuss the molecular chaperones associated with the receptor? Especially in lines 276-278.

  1. Line 37: conform is misused.

  1. Tandem affinity purification (TAP) was repeated several times, such as lines 83, 91, and 93.

  1. Lines 115 to 150 should be deleted?

  1. It is much better to reorient Fig. 1A (turn left 90 degrees), similar to Fig. 1 B orientation.

  1. Lines 372-373: may need to indicated as unpublished data.

  1. 8 can be placed side by side to save space.

Reviewer 3 Report

The paper by Knapp, Wolfrum, and collaborators addresses a member of

the family G-coupled protein receptors that is likely to be of high interest

to readers. Using affinity proteomics, the authors decipher molecular

mechanisms involving VLGR1/ADGRV in neuronal diseases beyond the  human Usher syndrom, i.e. in epilepsy and Alzheimer's.

The investigation methods are modern and appropriate and the description is quite detailed (though it could be simpler and more concise in places), therefore I fully support publication in Molecules.

I re-state briefly that
research on the pathways on which VLGPR1 acts is important for the community and that
the proposed involvement of the protein in epilepsy and Alzheimer's is an original, quite forward
proposition,  which also potentially triggers highly-rewarding further studies.
Through combined affinity proteomics and bioinformatics, the authors support the
conclusions above to a sufficient extent, in my opinion;  additional molecular interactions
on the proposed pathways could be analyzed, but such  investigations would fall outside the scope
of the current manuscript.
References seem appropriate.

additional point to be addressed:
- there are repetitions in the text of 'Materials and methods' at page 19.

Round 2

Reviewer 1 Report

I would like to thank the Authors for the detailed response to the comments.

Some minor comments are remained:

  1. Line 275 – astrocytes should be instead of astrocyte
  2. Line 295: fig 5c should be mentioned instead of 5d
  3. Description of method of mouse primary astrocyte cultivation is absent in the Materials and Methods section
  4. There are two versions of Figures 1,3,6 (old and new in some cases and duplicates of fig 6) and even three versions of fig 5. It can lead to confusion in published work.

Author Response

Response to reviewer 1´s comments and suggestions for the first revision:

I would like to thank the Authors for the detailed response to the comments.

We thank the reviewer for recognizing our efforts in the revision of our manuscript.

Some minor comments remain:

1. Line 275 – astrocytes should be instead of astrocyte

Response to 1: We thank the reviewer for pointing out this error and have changed "astrocytes" to the plural "astrocytes".

2. Line 295: fig 5c should be mentioned instead of 5d

Response to 2: The reviewer is correct and we followed the reviewer´s suggestion.

3. Description of method of mouse primary astrocyte cultivation is absent in the Materials and Methods section

Response to 3: We thank the reviewer for this comment. We missed adding the description for the cultivation of astrocytes and HeLa cells as well. We added “HEK293T, HeLa, and hTERT-RPE1 cells were cultured in Dulbecco’s modified Eagle’s medium (DMEM and DMEM-F12, respectively) containing 10% heat-inactivated fetal calf serum (FCS). Primary astrocytes were isolated from mouse brains and cultured as described in [117].” And cited our recent publication 117.        Güler, B.E.; Krzysko, J.; Wolfrum, U. Isolation and culturing of primary mouse astrocytes for the analysis of focal adhesion dynamics. STAR protocols 2021, 2, 100954, doi:10.1016/j.xpro.2021.100954.

In reviewing the references, we also noted that Berridge, 1983, was not included in the reference list. We have added the reference as follows:  20.  Berridge, M. J. Rapid accumulation of inositol trisphosphate reveals that agonists hydrolyse polyphosphoinositides instead of phosphatidylinositol. Biochemical journal 1983, 212, 849-858, doi:10.1042/bj2120849.

4. There are two versions of Figures 1,3,6 (old and new in some cases and duplicates of fig 6) and even three versions of fig 5. It can lead to confusion in published work.

Response to 4: We followed the instructions of the managing editor and left the deleted figures in the Word file of the manuscript as they were marked with the "Track Changes" function. In the 2nd revision of the manuscript, we have now accepted these changes and thus only the updated versions of the figures appear in the manuscript.